# Acacetin Protects against Non-Alcoholic Fatty Liver Disease by Regulating Lipid Accumulation and Inflammation in Mice

**DOI:** 10.3390/ijms23094687

**Published:** 2022-04-23

**Authors:** Chian-Jiun Liou, Shu-Ju Wu, Szu-Chuan Shen, Li-Chen Chen, Ya-Ling Chen, Wen-Chung Huang

**Affiliations:** 1Department of Nursing, Division of Basic Medical Sciences, Research Center for Chinese Herbal Medicine, Chang Gung University of Science and Technology, Taoyuan 33303, Taiwan; ccliu@mail.cgust.edu.tw; 2Division of Allergy, Asthma, and Rheumatology, Department of Pediatrics, Chang Gung Memorial Hospital, Linkou, Guishan Dist., Taoyuan 33303, Taiwan; lcchen@cgmh.org.tw; 3Department of Nutrition and Health Sciences, Research Center for Chinese Herbal Medicine, Chang Gung University of Science and Technology, Taoyuan 33303, Taiwan; sjwu@mail.cgust.edu.tw; 4Aesthetic Medical Center, Department of Dermatology, Chang Gung Memorial Hospital, Linkou, Taoyuan 33303, Taiwan; 5Graduate Program of Nutrition Science, National Taiwan Normal University, Taipei 11677, Taiwan; scs@ntnu.edu.tw; 6Department of Pediatrics, New Taipei Municipal TuCheng Hospital, Chang Gung Memorial Hospital, and Chang Gung University, New Taipei City 23652, Taiwan; 7School of Nutrition and Health Sciences, Taipei Medical University, Taipei 11031, Taiwan; 8Graduate Institute of Health Industry Technology, Research Center for Food and Cosmetic Safety, Chang Gung University of Science and Technology, Taoyuan 33303, Taiwan

**Keywords:** acacetin, inflammation, lipid metabolism, non-alcoholic fatty liver disease

## Abstract

We previously demonstrated that acacetin reduces adipogenesis in adipocytes, and decreases lipid accumulation in visceral adipocyte tissue. Here we investigated whether acacetin regulated the mechanisms of lipogenesis and inflammation in non-alcoholic fatty liver disease (NAFLD) in obese mice. Male C57BL/6 mice were fed a high-fat diet (HFD), and then administered acacetin by intraperitoneal injection. Acacetin reduced body weight and liver weight in obese mice. Acacetin-treated obese mice exhibited decreased lipid accumulation, increased glycogen accumulation, and improved hepatocyte steatosis. Acacetin regulated triglycerides and total cholesterol in the liver and serum. Acacetin decreased low-density lipoprotein and leptin concentrations, but increased high-density lipoprotein and adiponectin levels in obese mice. Acacetin effectively weakened the gene expressions of transcription factors related to lipogenesis, and promoted the expressions of genes related to lipolysis and fatty acid β-oxidation in liver. Acacetin also reduced expressions of inflammation-related cytokines in the serum and liver. Oleic acid induced lipid accumulation in murine FL83B hepatocytes, and the effects of acacetin treatment indicated that acacetin may regulate lipid metabolism through the AMPK pathway. Acacetin may protect against hepatic steatosis by modulating inflammation and AMPK expression.

## 1. Introduction

One important factor that contributes to obesity is the excessive intake of a high-calorie diet, which results in excessive lipid accumulation in visceral adipose tissue, as well as excessive triglyceride accumulation in hepatocytes, leading to hepatic steatosis formation [1]. Liver steatosis causes abnormal lipid and carbohydrate metabolism, and can result in the development and worsening of nonalcoholic fatty liver disease (NAFLD) [2]. If patients with NAFLD do not adjust their lifestyle and get a moderate amount of exercise, they are at risk of developing irreversible liver fibrosis, cirrhosis, hepatocellular carcinoma, and even death [3].

The pathogenesis of NAFLD is multifactorial, and has not been fully elucidated. Therefore, researchers continue to investigate the molecular mechanism and pathological development of NAFLD. The excessive intake of fat and carbohydrates by the digestive tract will increase the production of free fatty acids, which will be used to synthesize triglycerides in adipocytes or hepatocytes [4]. Fatty acids can also be synthesized from carbohydrates through acetyl-CoA, thus increasing the lipid accumulation in adipose and liver tissue [4]. Transcription factors for fatty acid synthesis, including CCAAT/enhancer-binding protein (C/EBP) and sterol regulatory element–binding protein 1c (SREBP-1c), will increase fatty acid synthase (FAS) expression and thereby enhance triglyceride production [5]. Recent studies show that increasing triglyceride breakdown in liver cells to produce free fatty acids is another strategy to reduce liver steatosis [6]. However, excessive free fatty acids will stimulate hepatocyte and adipocyte activation, causing more macrophage infiltration into the liver, thus leading to inflammation in hepatocytes. The activated macrophages will secrete more inflammatory cytokines, thereby stimulating insulin resistance, and promoting abnormal metabolism in adipocytes and hepatocytes [7]. Therefore, these free fatty acids must be decomposed through β-oxidation to produce energy and reduce the inflammation in liver tissue. Regulating lipogenesis, lipolysis, and fatty acid β-oxidation are important targets for improving lipid accumulation in NAFLD, and in indirectly alleviating hepatocyte damage caused by inflammation.

Acacetin is a flavone that has been isolated from various plants, including *Turnera diffusa* and *Saussurea involucrate* [8]. It was recently demonstrated that acacetin has anti-tumor effects and induces apoptosis in ovarian and breast cancer cells [9]. Acacetin could mitigate hypoxia-reoxygenation injury through increasing autophagy expression and the activation of the PI3K/Akt/mTOR pathway [10]. Previous studies also found that acacetin also improved cerebral ischemia-reperfusion injury by suppressing the NLRP3 pathway [11]. Acacetin attenuated inflammatory cytokine expressions in inflammatory macrophages, and reduced the expression of COX-2 and PGE_2_ [12]. Previous studies have found that acacetin can attenuate airway inflammation, airway hyperresponsiveness, and eosinophil infiltration in the lungs of asthmatic mice [13]. We previously demonstrated that acacetin can inhibit the lipid synthesis of 3T3-L1 adipocytes, and can reduce body weight and adipose tissue weight in obese mice by inhibiting lipogenesis [8]. However, it is not clear whether acacetin can ameliorate hepatic steatosis in obese mice. This study aimed to explore whether acacetin improved NAFLD in obese mice, and examined the lipogenesis and lipolysis pathways in this process.

## 2. Results

### 2.1. Acacetin Mitigated Body Weight in Obese Mice

Mice in the HFD group were fed an HFD for 16 weeks. The body weight gradually increased each week in the HFD group compared to the Normal group (Figure 1A). At the end of the animal experiments, mice with HFD-induced obesity and acacetin treatment (AC5 and AC10 groups) mitigated body weights compared to HFD group. Body weight was 41.03 ± 0.85 g in the AC5 group, and 38.51 ± 1.08 g in the AC10 groups, versus 45.79 ± 1.16 g in the HFD group (both *p* < 0.01; Figure 1A). We also measured the HFD-induced weight gain in mice given acacetin for 12 weeks. The acacetin-treated mice exhibited significantly less weight gain compared to the HFD group (Figure 1B). However, acacetin-treated mice fed a HFD did not exhibit decreased food intake compared to the HFD group (Figure 1C).

### 2.2. Acacetin Attenuated Liver Steatosis

Liver tissue weight effectively reduced in obese mice treated with 10 mg/kg acacetin (AC10 group) compared to obese mice (Figure 2A). However, obese mice treated with acacetin did not reduce the liver to body weight ratio compared to obese mice (Figure 2B). Liver tissue slices were stained to observe fat vacuoles. Obese mice increased hepatic macrophage aggregation and lipid vacuoles compared to the Normal group (Figure 2C). Acacetin treatment effectively suppressed the number of lipid vacuoles, the fat vacuole size, and macrophage aggregation in liver specimens compared to the HFD group (Figure 2C,D). Furthermore, acacetin-treated obese mice had decreased NAFLD scores compared to untreated obese mice (Figure 2E). Next, liver tissue slices were PAS stained to observe glycogen accumulation. Acacetin treatment increased the glycogen distribution in HFD-induced obese mice (Figure 3A). Glycogen measurement in liver tissues confirmed that acacetin treatment promoted glycogen levels in obese mice (Figure 3B). Acacetin also effectively inhibited TG and TC levels in the livers of obese mice (Figure 3C,D).

### 2.3. Acacetin Modulated Lipogenesis and Lipolysis in Liver Tissue

Compared to untreated obese mice, acacetin treatment significantly decreased the expressions of genes for transcription factors related to lipogenesis, including Srebp-1c, C/EBPα, and C/EBPβ (Figure 4A–C), and also decreased FAS gene expression (Figure 4D). Moreover, acacetin treatment increased expressions of the lipolysis-related genes adipose triglyceride lipase (ATGL) and hormone-sensitive lipase (HSL) (Figure 4E,F), and expressions of genes related to fatty acid β-oxidation, including peroxisome proliferator–activated receptor α (PPAR-α), carnitine palmitoyltransferase 1 (CPT-1), and carnitine palmitoyltransferase 2 (CPT-2) (Figure 4G–I). Furthermore, compared to the HFD group, the AC5 and AC10 groups exhibited significantly increased sirt1 gene expression (Figure 4J). Therefore, we examined protein expression in the liver, and found that acacetin-treated obese mice exhibited increased phosphorylated AMPK compared to obese mice (Figure 5A,B).

### 2.4. Acacetin Modulated Serum Metabolic Parameters

Analysis of serum metabolic parameters showed that acacetin attenuated the levels of TG, TC, LDL, and free fatty acids, and increased the HDL levels, compared to in untreated obese mice (Figure 6A–E). Acacetin administration reduced serum glucose and insulin levels compared to obese mice (Figure 7A,B). Moreover, acacetin-treated obese mice showed enhanced adiponectin production and reduced leptin expression in serum, compared to untreated obese mice (Figure 7C,D). We also examined whether acacetin showed liver toxicity in mice. The results demonstrated that acacetin-treated obese mice exhibited decreased serum levels of GOP and GPT compared to the HFD group (Figure 7E,F).

### 2.5. Acacetin Attenuated Liver Inflammation in Mice

We also examined the inflammation response in obese mice, and found that acacetin treatment led to significantly reduced TNF-α and IL-6 levels in the serum, compared to in untreated obese mice (Figure 8A,B). In liver tissue, the AC5 and AC10 groups also showed inhibited gene expressions of TNF-α and IL-6 compared to the HFD group, (Figure 8C,D).

### 2.6. Acacetin Attenuated Lipid Accumulation in FL83B Cells

Finally, we investigated whether acacetin treatment improved lipid metabolism in oleic acid-induced FL83B hepatocytes. Staining the cells with Oil Red O solution revealed that acacetin treatment reduced lipid droplet accumulation in oleic acid-induced FL83B cells (Figure 9A). Treatment of those cells with isopropanol revealed that 10–30 μM acacetin significantly reduced neutral lipid levels in oleic acid-induced FL83B cells (Figure 9B). Moreover, 30 μM acacetin exhibited significantly decreased FAS gene expression, and increased gene expression of ATGL and CPT-1 compared to oleic acid-induced FL83B cells. We also subjected oleic acid-induced FL83B cells to co-treatment with 30 μM acacetin and the AMPK inhibitor compound C, and found acacetin restored ATGL and CPT-1 gene expressions, and more effectively suppressed FAS gene expression (Figure 10).

## 3. Discussion

NAFLD is most commonly caused by obesity, mainly due to excessive fat accumulation in the liver [14]. We previously found that acacetin suppresses the lipid accumulation of 3T3-L1 adipocytes, and suppresses the lipid synthesis of adipocytes, mainly by reducing the expressions of transcription factors related to lipid synthesis, and thereby increasing triglyceride breakdown [8]. Acacetin also contributes to reducing lipid accumulation in the adipose tissue of obese mice [8]. Therefore, we speculated that acacetin should be able to reduce body weight, slow lipid accumulation in the liver, and ameliorate NAFLD in obese mice.

Being obese or overweight is a risk factor for several chronic diseases [15]. Clinical medical investigations have confirmed that overweight people exhibit excessive visceral fat accumulation, and increased lipoprotein deposition and plaque formation in the coronary artery, leading to atherosclerosis, hyperlipidemia, and other cardiovascular diseases [16]. Obesity is also directly related to the development of chronic diseases, such as diabetes and cancer [17]. Our previous research confirmed that acacetin can contribute to reducing lipogenesis and to decreasing the production of oil droplets in adipocytes. In our previous animal experiments, obese mice treated with acacetin exhibited a reduced visceral fat tissue weight [8]. Therefore, in our present study, acacetin treatment obviously reduced the weight of obese mice. Our present results also confirmed that acacetin could mitigate lipid accumulation in liver tissues and ameliorate NAFLD in obese mice.

There are two sources of TGs in serum. One source is the fat from food that is absorbed by the intestine. After digestion and absorption, these lipids are combined with protein to form chylomicrons, and enter the liver, adipose tissue, and other tissues through the circulatory system [18]. The other source of TGs is through synthesis by the liver and release into the blood [6]. Therefore, obese people often have a high serum concentration of TG. In our experiment, the obese mice showed high TG levels in both the serum and liver. Excessive TG accumulation in the liver is an important factor causing liver steatosis or NAFLD [14,19]. Interestingly, acacetin administration could reduce the TC levels in the liver and serum, and thereby also reduce the lipid vacuoles in the liver and the liver weight in obese mice. Mice fed a HFD induced obesity and accumulated large amounts of lipid in visceral and inguinal adipose tissue. HE staining of liver tissue demonstrated that obese mice had significantly increased fat vacuoles, and the number of lipid droplets compared to Normal group mice. In the liver, excessive lipid accumulation will interfere with energy metabolism, increasing the breakdown of glycogen to produce energy [20]. Therefore, HFD mice had significantly decreased glycogen accumulation compared to normal group mice. The density of lipids is lower than that of glycogen [21]. We found that liver weight increased by 1.22 times in obese mice compared with normal mice. HFD-induced obese mice had significantly increased 1.57 times body weight compared to the Normal group. Therefore, the ratio of liver weight to body weight decreased in obese mice compared to Normal group mice. Moreover, the liver weight increased by 1.16 and 1.20 times in obese mice compared with 5 mg/kg and 10 mg/kg for the acacetin group mice, respectively. We found that acacetin treatment increased the glycogen distribution in the liver of HFD-induced obese mice. However, the body weight of the obese group was 1.12 and 1.18 times that of the 5 mg/kg and 10 mg/kg acacetin group mice, respectively. Therefore, the ratio of liver weight to body weight was not effectively reduced in acacetin-treated obese mice.

Hepatocytes can increase FAS expression and accelerate fatty acid chain synthesis by promoting lipid synthesis transcription factors [20]. Acacetin effectively attenuated the gene expressions of C/EBPα, C/EBPβ, Srebp-1c, and FAS, confirming that acacetin decreased TGs in the liver by reducing lipid synthesis and thus improving liver steatosis in obese mice. Additionally, acacetin effectively reduced the oil droplet accumulation in FL83B liver cells. Cell experiments further confirmed that acacetin reduced FAS gene expression in oleic acid-induced FL83B cells. AMPK is an important regulating sensor for energy changes, and AMPK activation reduces the activation function of ACC [22]. Many studies have confirmed that the liver of obese mice exhibits significantly reduced AMPK expression, which inhibits ACC activation to block FAS production [19,23,24]. To confirm the importance of AMPK for lipid synthesis, AMPK inhibitor compound C evaluated lipogenesis in FL83B cells. We found that compound C increased the FAS expression in oleic acid-induced FL83B cells, while co-treatment with compound C and acacetin reduced FAS expression. Therefore, we suggest that acacetin may block lipid accumulation by promoting the expression of AMPK to inhibit lipogenesis in the liver of obese mice.

In obese patients, increasing the TG decomposition of fatty liver cells reduces the lipid accumulation in the liver [25]. This is also a strategy for improving liver steatosis and NAFLD. In the liver, ATGL cleaves TG to generate diglycerides and free fatty acids [25]. Additionally, diglycerides can be decomposed by activated HSL to produce free fatty acids and monoglycerides [25]. It has been previously demonstrated that some natural products, including resveratrol and maslinic acid, can increase ATGL and HSL expressions and thereby enhance the lipid decomposition of fatty liver in obese mice [26,27]. We previously found that acacetin could increase the ATGL and HSL expression in 3T3-L1 adipocytes. Moreover, acacetin treatment improves the ATGL and HSL expression in the adipose tissue of obese mice [8]. We confirmed that acacetin enhanced the gene expression of ATGL and HSL in the liver of obese mice, as well as the ATGL gene expression in oleic acid-induced FL83B cells. We also used the AMPK inhibitor compound C to evaluate how AMPK impacted the lipolysis of hepatocytes. Our results showed that compound C inhibited ATGL gene expression, and that co-treatment with compound C and acacetin restored ATGL gene expression in oleic acid-induced FL83B hepatocytes. These findings suggest that acacetin increased lipid breakdown by improving AMPK expression, and thereby reduced lipid accumulation in the liver of obese mice.

Triglyceride breakdown in the liver also produces excess free fatty acids. Fatty acids stimulate the inflammatory response of liver macrophages, which will also secrete inflammatory cytokines, causing persistent inflammation of fatty liver cells [28]. Activated macrophages secrete TNF-α to continuously stimulate liver cells, thereby causing insulin resistance of hepatocytes, and exacerbating the development of metabolic syndrome and diabetes in patients [29]. In our present experiments, acacetin treatment also decreased the gene expressions of IL-6 and TNF-α in the liver of obese mice. Hence, acacetin has the ability to reduce obesity-induced liver inflammation. However, our experiments did not investigate the distribution of M1 and M2 macrophages in the liver tissue. We also did not measure free fatty acid concentrations in the liver. Therefore, we speculated that acacetin not only increased lipid breakdown, but also resulted in the cleavage of free fatty acids to produce energy through fatty acid β-oxidation, thereby reducing the inflammation response due to excessive free fatty acids in the liver. Co-treatment with compound C and acacetin restored CPT-1 gene expression in oleic acid-induced FL83B hepatocytes. Therefore, our findings indicated that acacetin promoted AMPK activity to increase fatty acid β-oxidation and reduce liver inflammation.

NAFLD is the hepatic expression of the metabolic syndrome [30]. Our present experiments demonstrated that acacetin regulated fasting blood glucose and serum insulin concentration in obese mice. Treatment of obese mice with acacetin not only reduced serum and liver TC, but also lowered serum LDL and increased HDL levels, contributing to improving the development of metabolic syndrome. Notably, the adipocytes of obese mice secrete excessive leptin to block the hypothalamus and suppress appetite [31]. Obviously, as acacetin reduced serum leptin and increased the adiponectin concentration, it did not reduce body weight by reducing appetite. Hence, we thought that acacetin may reduce body weight by regulating lipid synthesis and accelerating lipolysis.

## 4. Materials and Methods

### 4.1. Animal Protocols

Acacetin (≥97.0% purity by HPLC) was purchased from Sigma-Aldrich (St. Louis, MO, USA). Four-week-old male C57BL/6 mice purchased from the National Laboratory Animal Center (Taipei, Taiwan). Thirty-six mice were separated into four groups (*n* = 8 per group): the normal control group (Normal); the high-fat diet group (HFD), in which mice were fed a diet containing 60% fat; and AC5 and AC10 groups, which included HFD-fed mice administered 5 or 10 mg/kg acacetin, respectively. Obesity was induced in mice as previously described [23]. First, the mice of the HFD, AC5, and AC10 groups were fed a HFD for four weeks. Subsequently, the AC5 and AC10 mice groups were continuously fed a HFD and treated with acacetin (dissolved in DMSO) by intraperitoneal injection twice per week for 12 weeks. Mice in the Normal and HFD groups received DMSO alone by intraperitoneal injection. Body weights were recorded weekly, and dietary intake was recorded daily, with food intake defined as weight of consumed food (g) × calorie of diet per day. The diet intake of mouse was monitored per day and body weight recorded weekly [32]. All animal experimental procedures were approved by the Laboratory Animal Care Committee of Chang Gung University of Science and Technology (IACUC approval number: 2019-008 and 2013-007). The HFD was based on the research diet TestDiet 58Y1 (Purina TestDiet, St. Louis, MO, USA). The HFD contains 60.9% of energy from fat (55.3% of energy from lard and 5.6% of energy from soyabean oil), 20.1% of energy from carbohydrates, and 18.3% of energy from protein [23].

### 4.2. Hepatic Histological Examination

Mice were anesthetized and euthanized. Liver tissue was removed and fixed with neutral buffered formalin solution for 24 h. Liver tissues were embedded in paraffin and cut into 6-μm sections. Subsequently, the slices were deparaffinized and stained using hematoxylin and eosin (HE) solution, as previously described [23]. Briefly, the slide was stained with haematoxylin solution. Next, the slide was washed, and eosin solution was added for 30 s at room temperature. Subsequently, the biopsy specimens were dehydrated and observed using a light microscope (Olympus, Tokyo, Japan). The three parameters of steatosis, lobular inflammation (macrophage aggregation), and hepatocyte ballooning was determined using the NAFLD score as previously described [33]. The score was calculated as the unweighted sum of the scores for steatosis (0–3), lobular inflammation (0–3), and ballooning (0–2), and ranges from 0 to 8. Briefly, steatosis was graded according to the percentage of total fat area in the liver biopsy: 0 (<5%), 1 (5–33%), 2 (34–66%), and 3 (>66% of hepatocytes had fatty changes). The score of inflammatory cells (macrophage) was calculated in the same region. Inflammatory cell scores included 0 (none), 1 (<2 foci per × 200 field), 2 (2–4 foci per × 200 field) and 3 (>4 foci per × 200 field). The score of ballooning included 0 (none), 1 (few ballooning cells), and 2 (many ballooning cells) [33]. Furthermore, we observed glycogen accumulation in the liver tissues using periodic acid-Schiff (PAS) staining. For the first step of PAS staining, the slide was deparaffinized and treated with periodic acid solution for 5 min. Next, the slide was washed, and Schiff’s reagent was added for 15 min at room temperature. Subsequently, hematoxylin solution was added, and the biopsy specimens were observed using a light microscope (Olympus).

### 4.3. Biochemical Analysis

At the end of the animal experiments, mice were anesthetized and euthanised with 4% isoflurane. Blood was taken from the orbital vascular plexus and centrifuged at 6000 rpm to obtain serum. The serum samples were analyzed for levels of glutamic oxaloacetic transaminase (GOP), glutamic pyruvic transaminase (GPT), total cholesterol (TC), high-density lipoprotein (HDL), and total triglycerides (TGs) using a biochemical analyzer (DRI-CHEM NX500; Fuji, Tokyo, Japan). Low-density lipoprotein (LDL) was assayed using the LDL assay kit (Sigma). Free fatty acids were detected using a fatty acid quantitation kit (Sigma), according to the manufacturer’s instructions. Briefly, we prepared a dilution series of standard in the concentration range of 1–1000 μM. Standard and serum sample added ACS reagent for 30 min at 37 °C. Subsequently, each well was treated with a master reaction mix reagent for 30 min at 37 °C. The levels of free fatty acids were detected at an optical density (OD) of 570 nm using a microplate reader (Thermo, Waltham, MA, USA). Furthermore, the day before the mice were sacrificed, they were fasted for 16 h. Following this fasting period, all mice were administered glucose by intraperitoneal injection. Blood sugar level detected using the biochemical analyzer. Serum insulin concentration was examined by the Insulin EIA Kit (Cayman, MS, USA). Furthermore, liver glycogen concentration was detected by a Glycogen Assay Kit (Cayman, MS, USA), according to the manufacturer’s protocol. Liver glycogen levels were detected by absorbance at 570 nm using a microplate reader (Thermo).

### 4.4. Cell Culture and Treatments

The Murine FL83B hepatocyte cell line was purchased from the Bioresource Collection and Research Center (Taiwan). Cells were cultured in F-12K medium (Invitrogen-Gibco, Paisley, Scotland) containing 10% fetal bovine serum. To induce lipid accumulation in hepatocytes, FL83B cells were seeded on culture plates and induced with 0.5 mM oleic acid for 48 h. Subsequently, cells were treated with 0.1% DMSO or 10 or 30 μM acacetin for 24 h to observe lipid metabolism. Additionally, oleic acid-induced FL83B cells were co-treated with 30 μM acacetin and the AMPK inhibitor compound C (10 μM; Sigma), and we assayed the expressions of genes related to the lipid metabolism.

### 4.5. Oil Red O Staining

FL83B hepatocytes were seeded on culture plates and induced with 0.5 mM oleic acid for 48 h. Next, cells were treated with 10 or 30 μM acacetin for 24 h. Cells were fixed using formalin, and then stained used Oil Red O solution to observe neutral lipids by light microscope (Olympus). Finally, the cell culture plates were treated with isopropanol, and lipid accumulation was determined by absorbance at 490 nm using a microplate reader (Thermo).

### 4.6. Real-Time PCR

Total RNA was extracted from FL83B hepatocytes or liver tissues using TRIzol reagent solution (Sigma). cDNA was synthesized using a cDNA synthesis kit (Thermo). Next, an SYBR green master mix kit (Bio-Rad, San Francisco, CA, USA) was used to amplify specific genes using a spectrofluorometric thermal cycler (Bio-Rad). The sequences of primer were presented in Table 1 [23,34]. The average of the gene cycle threshold (Ct) was measured for each experiment. Fold changes were calculated relative to β-actin expression.

### 4.7. Western Blot

Liver tissues homogenized and extracted protein by protein lysis buffer (50 mM Tris–HCl, pH8, 150 mM NaCl, 0.5% NP40, 0.1% SDS, 1 mM EDTA) containing protein inhibitor cocktail (Sigma). Total proteins were separated by SDS-PAGE and transferred to PVDF membrane. The membrane was then incubated with specific primary antibodies overnight. Next, the PVDF membrane was incubated with secondary antibodies (diluted at 1:5000). Protein bands were detected using chemiluminescent substrate (Thermo), and observed and quantified by a BioSpectrum protein imaging system (UVP, Upland, CA, USA). We utilized primary antibodies against AMPKα (diluted at 1:1000), phosphorylated AMPKα (pAMPKα) (diluted at 1:1000) (Cell Signaling Technology, Danvers, MA, USA), and β-actin (diluted at 1:10,000) (Sigma).

### 4.8. Data Analysis

Statistical analyses were performed with SPSS v19 (SPSS, Chicago, IL, USA). Animal experiment results were analyzed by one-way analysis of variance (ANOVA), followed by the Dunnett and post hoc test for multiple comparisons. Between the two groups, an unpaired Student’s *t*-test was used in the cell experiment. Statistical significance was set at *p* < 0.05, and the results were expressed as the mean ±SEM. All experiments were performed at least three times.

## 5. Conclusions

In conclusion, our present experiments confirmed that acacetin could regulate the lipid metabolism of NAFLD by improving AMPK expression. The above studies showed that acacetin may have potential as a natural compound for NAFLD treatment.

## Figures and Tables

**Figure 1 ijms-23-04687-f001:**
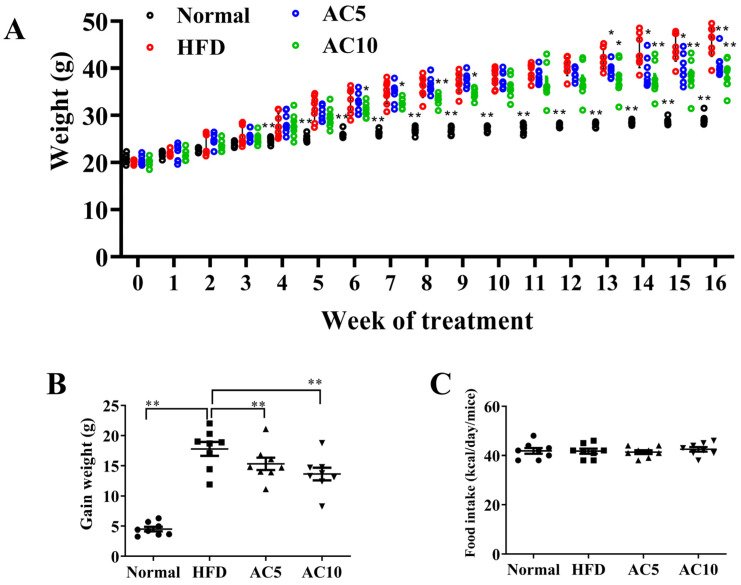
The biometrics of HFD-induced obese mice treated with acacetin (AC). (**A**) Body weight recorded weekly for 16 weeks. (**B**) The weight gain of HFD-induced obese mice that were treated with acacetin from week five to week 16. (**C**) Food intake recorded daily. The values are presented as mean ± SEM (*n* = 8 per group). * *p* < 0.05, ** *p* < 0.01 compared to HFD group.

**Figure 2 ijms-23-04687-f002:**
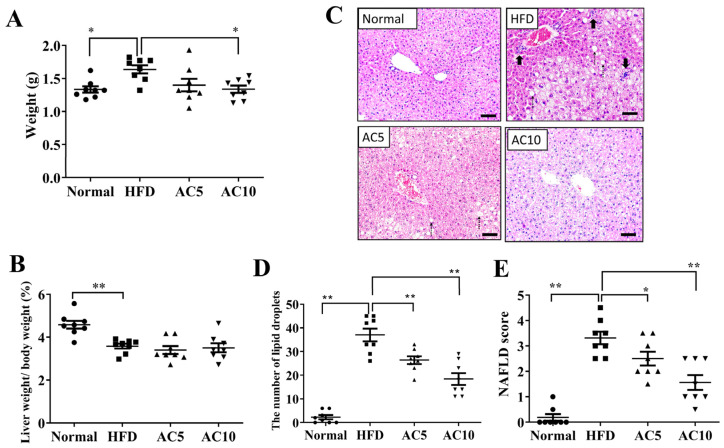
Biometrics of NAFLD in mice treated with acacetin (AC). (**A**) Liver weight. (**B**) the ratio of liver weight to body weight. (**C**) HE staining of liver tissues (200× magnification). (**D**) Number of lipid vacuoles in liver tissues. (**E**) NAFLD scores in liver tissues. Values are presented as mean ± SEM (*n* = 8 per group). * *p* < 0.05, ** *p* < 0.01 indicate significant differences compared to obese mice. (scale bar = 100 µm; bold arrow: lobular inflammation (macrophage aggregation); dotted line arrow: ballooning).

**Figure 3 ijms-23-04687-f003:**
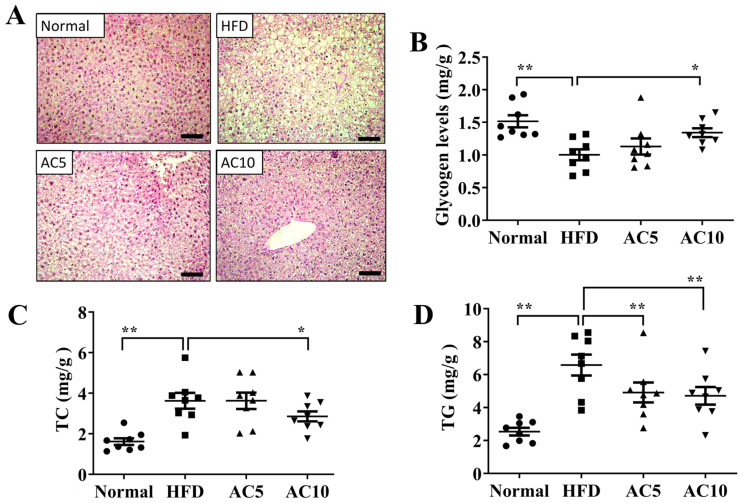
Effect of acacetin (AC) on histopathology of NAFLD in mice. (**A**) PAS staining revealed glycogen distribution in liver (200× magnification). Quantification of (**B**) glycogen, (**C**) TC, and (**D**) TG levels in liver tissue. Values are presented as mean ± SEM. * *p* < 0.05, ** *p* < 0.01 compared to obese mice. (scale bar = 100 µm).

**Figure 4 ijms-23-04687-f004:**
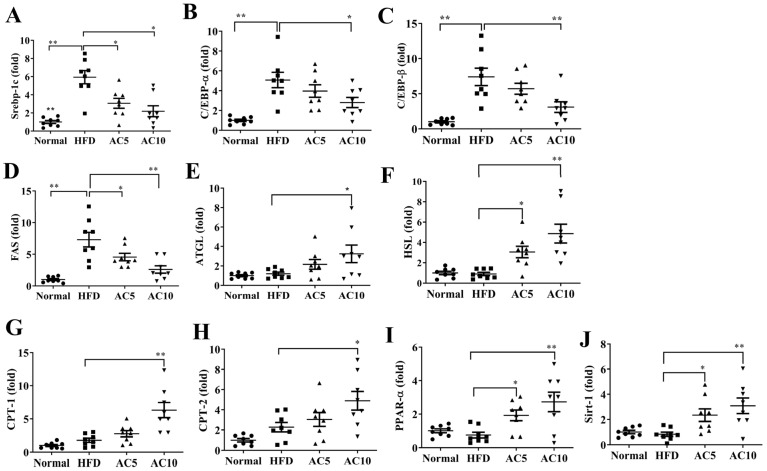
Acacetin (AC) modulated gene expression in liver tissue. Real-time RT-PCR detected the gene expression of (**A**) Srebp-1C, (**B**) C/EBPα, (**C**) C/EBPβ, (**D**) FAS, (**E**) ATGL, (**F**) HSL, (**G**) CPT-1, (**H**) CPT-2, (**I**) PPAR-α, and (**J**) Sirt-1. The fold values were measured relative to β-actin expression. Values are presented as mean ± SEM. * *p* < 0.05, ** *p* < 0.01 compared to obese mice.

**Figure 5 ijms-23-04687-f005:**
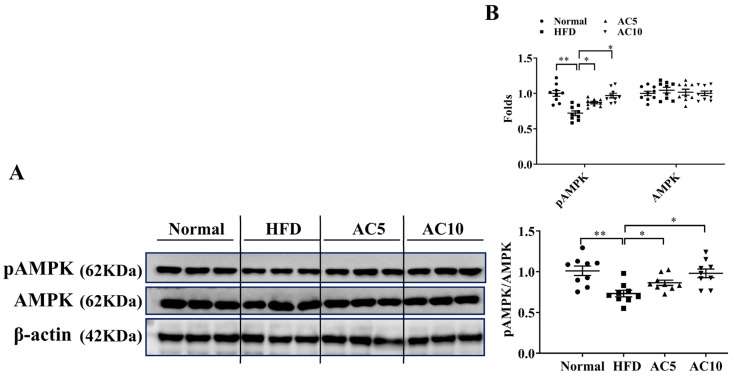
Acacetin (AC) regulated AMPK expression in the liver. (**A**) pAMPK and AMPK protein expressions. (**B**) The fold values were calculated relative to β-actin expression, and the ratio of pAMPK/AMPK. Values are presented as mean ± SEM of three independent experiments. * *p* < 0.05, ** *p* < 0.01 compared to obese mice.

**Figure 6 ijms-23-04687-f006:**
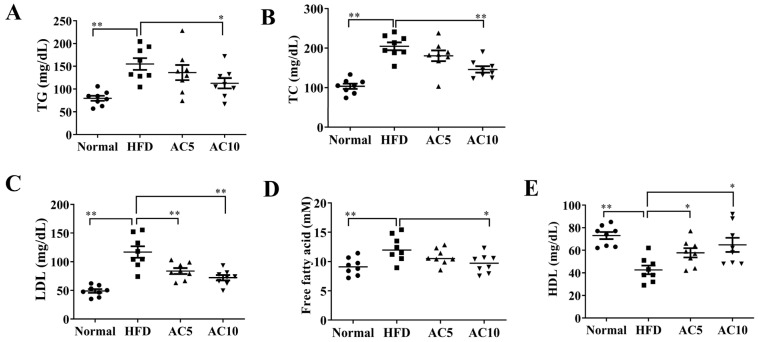
Effects of acacetin (AC) on serum TG and TC in mice. Levels of (**A**) TG, (**B**) TC, (**C**) LDL, (**D**) free fatty acids, and (**E**) HDL. Values are presented as mean ± SEM. * *p* < 0.05, ** *p* < 0.01 indicate significant differences compared to obese mice.

**Figure 7 ijms-23-04687-f007:**
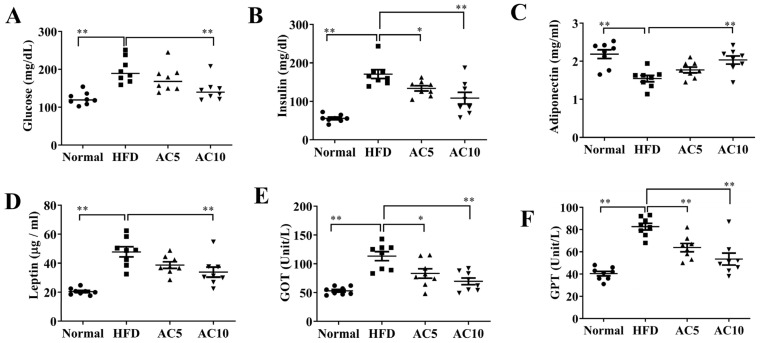
Serum biochemical analysis in mice. Levels of (**A**) glucose, (**B**) insulin, (**C**) adiponectin, (**D**) leptin, (**E**) GOT, and (**F**) GPT. Values are presented as mean ± SEM. * *p* < 0.05, ** *p* < 0.01 indicate significant differences compared to obese mice.

**Figure 8 ijms-23-04687-f008:**
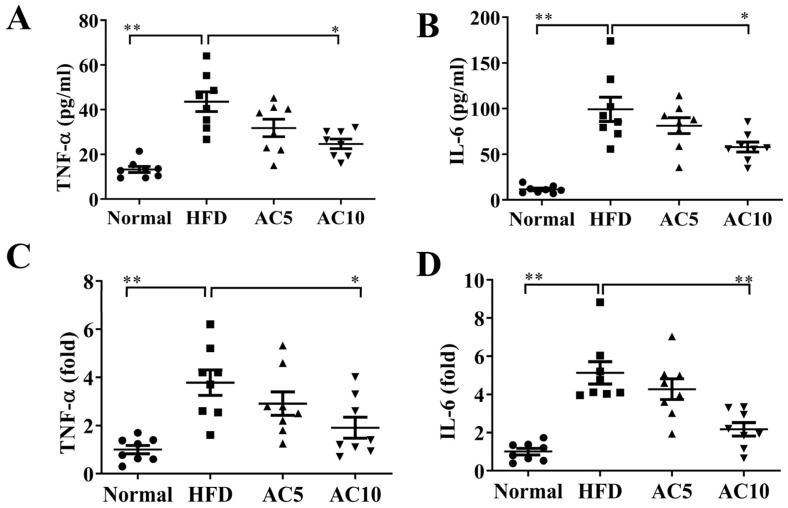
Acacetin (AC) reduced inflammatory response in mice. AC inhibited (**A**) TNF-α and (**B**) IL-6 levels in serum. (**C**) TNF-α and (**D**) IL-6 gene expression in liver tissue. The fold values were measured relative to β-actin expression. Values are presented as mean ± SEM. * *p* < 0.05, ** *p* < 0.01 compared to obese mice.

**Figure 9 ijms-23-04687-f009:**
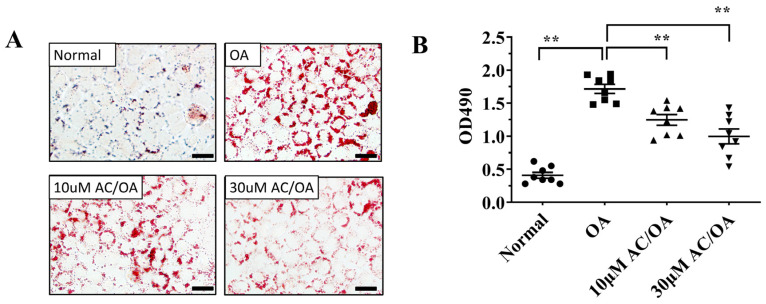
Acacetin (AC) attenuated lipid accumulation in FL83B cells. (**A**) Oil Red O staining revealed lipid accumulation (200× magnification). (**B**) FL83B cells treated with isopropanol and lipid accumulation was measured using the absorbance at OD 490 nm. Values are presented as mean ± SEM. ** *p* < 0.01 indicate significant differences compared to FL83B cells treated with oleic acid (OA). (scale bar = 100 µm).

**Figure 10 ijms-23-04687-f010:**
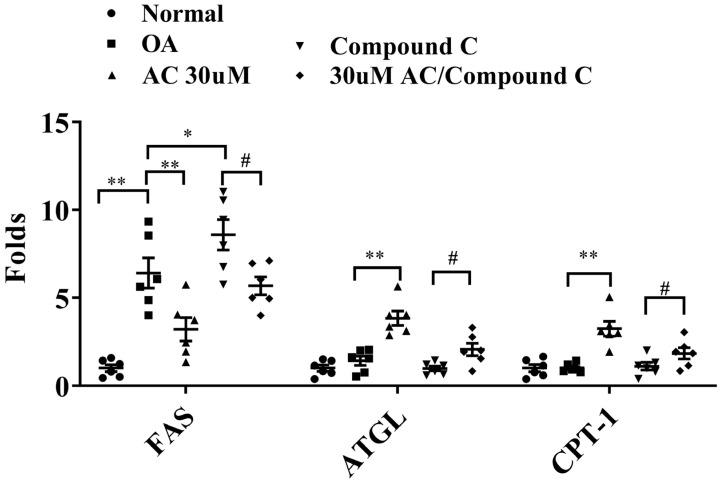
Effects of acacetin (AC) on the AMPK pathway in FL83B cells. FL83B cells treated with 0.5 mM oleic acid (OA) for 48 h, followed by 24 h of treatment with 30 μM AC with or without an AMPK inhibitor (compound c). Gene expression levels of FAS, ATGL, and CPT-1 were assayed using real-time PCR. The fold values calculated relative to β-actin expression. Values are presented as mean ± SEM. * *p* < 0.05, ** *p* < 0.01 indicate significant differences compared to OA treatment. # *p* < 0.05 indicate significant differences compared to compound c treatment.

**Table 1 ijms-23-04687-t001:** Sequences of primer pairs used for real-time PCR. Forward (F); Reverse (R).

Gene	Primer	5′–3′ Sequence
ATGL	F	CTCAGGCGAGAGTGACATCT
	R	GATTGCGAAGGTTGAACTGGAT
C/EBPα	F	TGGAGACGCAACAGAAGG
	R	TGTCCAGTTCACGGCTCA
C/EBPβ	F	GTCCAAACCAACCGCACAT
	R	CAGAGGGAGAAGCAGAGAGTT
CPT1	F	GAGCCAGACCTTGAAGTAACG
CPT2	R	GAGACAGACACCATCCAACAC
	F	TTGACCAGTGAGAACCGAGAT
	R	AGAGGCAGAAGACAGCAGAG
FAS	F	ATCCTGGCTGACGAAGACTC
	R	TGCTGCTGAGGTTGGAGAG
HSL	F	CGGCGGCTGTCTAATGTCT
	R	CGTTGGCTGGTGTCTCTGT
PPAR-α	F	GGAGCGTTGTCTGGAGGTT
	R	GAAGTGGTGGCTAAGTTGTTGA
Sirt1	F	CGTCTTGTCCTCTAGTTCCTGT
	R	GCCTCTCCGTATCATCTTCCA
SREBP-1c	F	CTGTTGGTGCTCGTCTCCT
	R	TTGCGATGCCTCCAGAAGTA
β-actin	F	AAGACCTCTATGCCAACACAGT
	R	AGCCAGAGCAGTAATCTCCTTC

## Data Availability

The data presented in this study are available on request from the corresponding author.

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
