# Peer review of "Acacetin Protects against Non-Alcoholic Fatty Liver Disease by Regulating Lipid Accumulation and Inflammation in Mice"

_ijms, 2022, doi:10.3390/ijms23094687_

Round 1

Reviewer 1 Report

The manuscript “Acacetin protects against non-alcoholic fatty liver disease by regulating lipid accumulation and inflammation in mice” by Chian-Jiun Liou et al. They have; reported that acacetin could play an important role in the regulation of the lipid metabolism of nonalcoholic fatty liver disease (NAFLD) by improving AMPK expression. The manuscript is written in standard English with few grammatical errors. Based on the current status of this manuscript, I think a minor revision is necessary.

Comments:

  1. The authors should check grammatical errors very carefully before resubmission.
  2. I will suggest authors add a few more sentences about the beneficial properties of Acacetin in the introduction section (line no: 72)
  3. What rationale is behind 5 or 10 mg/kg acacetin dose selection?
  4. In the figure, the 3A scale bar is missing. I will suggest adding a scale bar in the figures
  5. In figure 5A, western blot images, the author should add the molecular weight of the specific proteins
  6. In the figure, the 9A scale bar is missing,
  7. Section 2.7 western blot analysis specifies the buffer used to extract proteins.
  8. Specify the dilutions of primary and secondary antibodies in section 2.7
  9. I will suggest adding a graphical abstract, which is the point of attraction for a reader.

Author Response

The manuscript “Acacetin protects against non-alcoholic fatty liver disease by regulating lipid accumulation and inflammation in mice” by Chian-Jiun Liou et al. They have; reported that acacetin could play an important role in the regulation of the lipid metabolism of nonalcoholic fatty liver disease (NAFLD) by improving AMPK expression. The manuscript is written in standard English with few grammatical errors. Based on the current status of this manuscript, I think a minor revision is necessary.

Comments:

1.The authors should check grammatical errors very carefully before resubmission.

Responses:

Thank for reviewer’s suggestion. We checked the manuscript by San Francisco Edit (Invoice No: 22002324).

2.I will suggest authors add a few more sentences about the beneficial properties of Acacetin in the introduction section (line no: 72)

Responses:

Thank for reviewer’s suggestion. We add more information about acetin research in Line 73-83

3.What rationale is behind 5 or 10 mg/kg acacetin dose selection?

Responses:

In the present study, we found that mice could tolerated 7 daily administrations of 50 mg/kg, 20 mg/kg and 10mg/kg acacetin (results not shown). However, 50 mg/kg acacetin was judged too high for a clinical dose. We found that 10 mg/kg acacetin could reduce the serum levels of glutamate oxaloacetate transaminase and glutamic pyruvic transaminase, recovering liver function in mice. Therefore, we used 10 and 5 mg/kg acacetin in animal experiments to explore the effects of acacetin on NAFLD pathology in obese mice.

4.In the figure, the 3A scale bar is missing. I will suggest adding a scale bar in the figures

Responses:

Thank for reviewer’s suggestion. We added scale bar in the figures 3A

5.In figure 5A, western blot images, the author should add the molecular weight of the specific proteins

Responses:

Thank for reviewer’s suggestion. We added s the molecular weight of the specific proteins in the figures 5A

6.In the figure, the 9A scale bar is missing,

Responses:

Thank for reviewer’s suggestion. We added scale bar in the figures 9A

7.Section 2.7 western blot analysis specifies the buffer used to extract proteins.

Responses:

We added more description about protein lysis buffer in line 397-399.

“Liver tissues homogenized and extracted protein by protein lysis buffer (50 mM Tris–HCl, pH8, 150 mM NaCl, 0.5% NP40, 0.1% SDS, 1 mM EDTA) containing protein inhibitor cocktail (Sigma).”

8.Specify the dilutions of primary and secondary antibodies in section 2.7

Responses:

 We added the dilutions of primary and secondary antibodies in line 401-405.

9.I will suggest adding a graphical abstract, which is the point of attraction for a reader.

Responses:

We added a graphical abstract. Thank you again for your positive comments and valuable suggestions to improve the quality of our manuscript.

Reviewer 2 Report

In this article the authors reported the protective effect of Acacetin against NAFLD; this flavonoid regulates inflammation and lipid accumulation via the AMPK pathway

Major comments

Literature evidence of recent years, reveals that natural compound, such as flavonoid, are potential candidates for ameliorating NAFLD.

It's well known that flavonoids have therapeutic effects on obesity and NAFLD through immune-modulatory, anti-inflammatory and antioxidant properties. Similarly, it is also well known that flavonoids are capable of attenuating lipid accumulation in vivo and in vitro reducing the expression of the key transcriptional factors and lipogenic enzymes such as SREBP-1c, PPAR-g, FAS and ACC. In addition, AMPK phosphorylation have been mentioned frequently in hepatic lipid metabolism to be activated in response to many natural compound suggesting that flavonoids may serve as a potential therapeutic agent for the treatment of NAFLD.

In my opinion this work is unoriginal and does not provide innovative data.

Some preclinical studies have shown that natural compounds such as flavonoids offer therapeutic potential through the regulation of FOXO 1 (forkhead box O1) and associated pathways.

Haven't the authors thought of evaluating this transcriptional factor?

Minor comments

Methods

-To better clarify authors should describe the methods rather than limit themselves to self-citations

- The statistical package used for the analysis is missing in the data analysis section

Results

Authors should review and make improvements to the figures reported:

-Please, individual data point should be shown and not dynamite plots

- Please, for more clarity authors should specify what is reported on the y-axis in both Figure 1B and Figure 2A

-Histological panels should be enlarged for better visibility and authors should reported scale bar also in fig.3 and fig. 9

Author Response

Major comments

Literature evidence of recent years, reveals that natural compound, such as flavonoid, are potential candidates for ameliorating NAFLD.

It's well known that flavonoids have therapeutic effects on obesity and NAFLD through immune-modulatory, anti-inflammatory and antioxidant properties. Similarly, it is also well known that flavonoids are capable of attenuating lipid accumulation in vivo and in vitro reducing the expression of the key transcriptional factors and lipogenic enzymes such as SREBP-1c, PPAR-g, FAS and ACC. In addition, AMPK phosphorylation have been mentioned frequently in hepatic lipid metabolism to be activated in response to many natural compound suggesting that flavonoids may serve as a potential therapeutic agent for the treatment of NAFLD.

In my opinion this work is unoriginal and does not provide innovative data.

Some preclinical studies have shown that natural compounds such as flavonoidors offer therapeutic potential through the regulation of FOXO 1 (forkhead box O1) and associated pathways. Haven't the authors thought of evaluating this transcriptional factor?

Responses:

Some flavonoids could improve NAFLD in obese mice thought reduced lipogenesis and promote lipolysis. Many studies have found that purified compounds or flavonoids of plants may improve NAFLD. For example, 6-gingerol is linked to reduced body weight and inhibition of lipid accumulation of hepatocytes in obese mice. Celastrol has been associated with increased Sirt-1 expression, suggesting improved liver lipid metabolism, and with reduced liver damage in HFD induced obese mice. Curcumin and astaxanthin can inhibit liver inflammation and fibrosis and decrease lipid and adipose tissue accumulation. Albiflorin also significantly reduced the weights of liver and white adipose tissue.

   AMPK is an energy sensor involved in a pathway that largely regulates energy storage and consumption in liver and white adipose tissue. The cells accumulate excess energy, leading to enhanced AMPK phosphorylation and activity for induced ACC phosphorylateion. Many studies have found that ACC phosphorylation will decrease and block downstream FAS expression associated with the synthesis of fatty acid chains. Recent reports also describe Sirt-1 as regulating AMPK phosphorylation, leading to increased AMPK activity. Therefore, enhancing the Sirt-1/AMPK pathway would be expected to block fatty acid chain synthesis and reduce hepatocyte lipid accumulation.

   Our laboratory mainly studies natural products to improve obesity and non-alcoholic fatty liver disease. We also investigated the molecular mechanisms of lipid metabolism and immune inflammation in adipose tissue and fatty liver of obese mice. Previously, our study found that licochalcone A and maslinic acid could ameliorate obesity and NAFLD via promotion of the Sirt-1/AMPK pathway in obese mice. We also found that phloretin, is an apple polyphenol, could ameliorate hepatic steatosis through regulation of lipogenesis and Sirt1/AMPK signaling in obese mice. Therefore, our research focused on the molecular mechanism by which the sirt1/AMPK pathway regulates lipid metabolism in NAFLD of obese mice.

   Recent studies have found that FoxO1 is an important transcription factor involved in energy metabolism [1]. In hepatocytes, FOXO1 can bind to the promoter of Apolipoprotein C-III gene for promoting its transcription, and elevating the triglyceride levels in plasma and liver [2]. In addition, other studies have found that inhibition of FoxO1-dependent inhibition of glucokinase can also block lipogenesis in NAFLD of obese mice [3].

   In this study, our experiment demonstrated that acacetin could reduce the body weight and the levels of triglycerides in serum and liver tissue of obese mice. Acacetin could also regulate lipid metabolism by increasing the Sirt1/AMPK pathway, improving liver steatosis in obese mice. However, our study did not investigate the regulation of lipid metabolism pathways by FOXO-1 pathway. We are grateful to the reviewer for providing novel information on the molecular mechanisms of metabolic regulation. In the future, we will also investigate the molecular pathway of FOXO1 for regulated the molecular mechanism of lipid metabolism to improve of NAFLD.

References

  1. Peng, S.; Li, W.; Hou, N.; Huang, N. A review of foxo1-regulated metabolic diseases and related drug discoveries. Cells 2020, 9, 184.
  2. Langlet, F.; Haeusler, R.A.; Lindén, D.; Ericson, E.; Norris, T.; Johansson, A.; Cook, J.R.; Aizawa, K.; Wang, L.; Buettner, C., et al. Selective inhibition of foxo1 activator/repressor balance modulates hepatic glucose handling. Cell 2017, 171, 824-835.e818.
  3. Yang, Z.; Roth, K.; Agarwal, M.; Liu, W.; Petriello, M.C. The transcription factors crebh, ppara, and foxo1 as critical hepatic mediators of diet-induced metabolic dysregulation. J Nutr Biochem 2021, 95, 108633.

Minor comments

Methods

-To better clarify authors should describe the methods rather than limit themselves to self-citations

Responses:

Thank for reviewer’s suggestion. We modify and add more experimental description in Methods.

- The statistical package used for the analysis is missing in the data analysis section

Responses:

We modify the description of statistical analysis as

“Statistical analyses were performed with SPSS v19 (SPSS, Chicago, IL, USA). Animal experiment results were analyzed by one-way analysis of variance (ANOVA), followed by the Dunnett and post hoc test for multiple comparisons. Between two groups, an unpaired Student t-test was used in cell experiment. Statistical significance was set at P < 0.05, and the results were expressed as the mean ±SEM. All experiments were performed at least three independent experiments.”

Results

Authors should review and make improvements to the figures reported:

-Please, individual data point should be shown and not dynamite plots

Responses:

Thank for reviewer’s suggestion. We modified Figure 1A as data point.

- Please, for more clarity authors should specify what is reported on the y-axis in both Figure 1B and Figure 2A

Responses:

Thank for reviewer’s suggestion. We modified the y-axis in Figure 1B and Figure 2A.

-Histological panels should be enlarged for better visibility and authors should reported scale bar also in fig.3 and fig. 9

Responses:

Thank for reviewer’s suggestion. We added scale bar in the figures 3A and Figure 9A

Round 2

Reviewer 2 Report

As request in the first revision authors should review and make improvements to the all figures reported. Dot plot instead of plunger plot (bar graphs) should be shown  in all figures. The dot plot shows each observation  sampled. On the other hand, the plunger plot obscures both the number of values and their distribution. 

Author Response

As request in the first revision authors should review and make improvements to the all figures reported. Dot plot instead of plunger plot (bar graphs) should be shown in all figures. The dot plot shows each observation sampled. On the other hand, the plunger plot obscures both the number of values and their distribution.

Responses:

We modified as dot plot figure in all plunger plot figure. Thank you again for your positive comments and valuable suggestions to improve the quality of our manuscript.
